# Effect of Jasmonic Acid, Yeast Extract Elicitation, and Drying Methods on the Main Bioactive Compounds and Consumer Quality of Lovage (*Levisticum officinale* Koch)

**DOI:** 10.3390/foods9030323

**Published:** 2020-03-11

**Authors:** Urszula Złotek, Urszula Szymanowska, Kamila Rybczyńska-Tkaczyk, Anna Jakubczyk

**Affiliations:** 1Department of Biochemistry and Food Chemistry, University of Life Sciences in Lublin, Skromna Str. 8, 20-704 Lublin, Poland; urszula.szymanowska@up.lublin.pl (U.S.); anna.jakubczyk@up.lublin.pl (A.J.); 2Department of Environmental Microbiology, Laboratory of Mycology, The University of Life Sciences in Lublin, Leszczyńskiego Street 7, 20-069 Lublin, Poland; kamila.rybczynska-tkaczyk@up.lublin.pl

**Keywords:** lovage, elicitation, bioactive compounds, color, consumer acceptability

## Abstract

The aim of the study was to evaluate changes in the activities of some enzymes (polyphenol oxidase—PPO and peroxidase—POD), the content of some bioactive compounds, and the organoleptic quality and color parameters of fresh lovage and its herb dried with various methods and elicited with jasmonic acid (JA) and yeast extract (YE). Elicitation only slightly affected the sensory quality of the fresh herbs, but consumer responses in terms of acceptability of the dried lovage color showed that lovage from microwave drying was least acceptable. The largest increase in the value of parameter a* was observed in microwave dried samples. Elicitation positively influenced the content of bioactive compounds (especially chlorophylls, vitamin C, and phenolic compounds), but unfortunately drying caused significant loss of bioactive compounds (except phenolic compounds) in both control and elicited samples. Drying also resulted in a decrease in the activity of PPO and POD.

## 1. Introduction

The quality of herbs is connected with two main aspects: Visual and organoleptic quality and pro-health properties. Nowadays, consumers devote increasing attention to the bioactivity of spices and herbs, but the first impression associated with organoleptic properties still largely determines the acceptability of these plant products. Therefore, consumer quality of herbs should be associated with both these aspects. Lovage, which is a culinary and medicinal herb, possesses many biological activities, e.g., antioxidant, antibacterial, hepatoprotective, vasorelaxant, cyclooxygenase inhibitory, and antitumor effects [1].

Jasmonic acid belongs to the group of plant hormones called “jasmonates.” It is a lipid-derived compound synthesized via the octadecanoid pathway. This plant hormone influences some physiological plant processes, e.g., development, structure, and flowering. Additionally, JA plays an important role in signaling in plants and is especially connected with the acquisition of systemic resistance via jasmonate-dependent signaling pathways [2].

Yeast extract is a very interesting natural plant inducer due to its role in producing some growth regulators, as well as its ability to act as a biostimulant of plant growth or the biosynthesis of plant pigments and some other bioactive compounds [3,4]. For example, in a study conducted by Zlotek et al. [5], yeast extract (0.1% and 1%) elicitation increased the level of phenolic compounds and chlorophylls in lettuce leaves. In turn, in marjoram leaves, this elicitor caused an increase in the content of ascorbic acid and chlorophylls [4]. The foliar application of 5% to 20% yeast extract was found to increase the contents of chlorophylls, carotenoids, and phenolic compounds in neem leaves [3].

As indicated in our previous study, elicitation with jasmonic acid and yeast extract can improve some biological activities via enhancement of the production of phenolic compounds [6]. Elicitation can also influence the biosynthesis of other groups of bioactive compounds like vitamins and plant pigments (chlorophylls and carotenoids), which can influence the organoleptic quality of plants, especially their color. Additionally, diets rich in vitamins and carotenoids have also been epidemiologically correlated with a lower risk of several diseases [7]. It has been well-documented that elicitation can influence the activity of plant defense enzymes such as peroxidase (POD) or polyphenol oxidase (PPO). These enzymes play an important role in plant resistance but may also influence the consumer quality of plant food. PPO and POD are the main enzymes involved in the enzymatic browning reaction in plants and may be responsible for deterioration of the appearance of food of plant origin [8]. PPO activity may also determine the flavor and aroma of plant products, since some phenolic compounds (substrates for PPO) play a role in taste modulation. In addition, PPO is believed to be responsible for oxidative degradation of ascorbic acid [8]. Another enzyme that is largely believed to be involved in color and flavor degradation of plant food is peroxidase. POD causes oxidation of phenolic compounds in the presence of hydrogen peroxide and formation of brown degradation products. There are also some documents indicating that increased POD activity may be correlated with chlorophyll degradation in the green parts of plants. On the positive side, POD activity can improve the textural quality of some greens during thermal processing via catalyzing the formation of cross-linking polymers of the plant cell wall [8].

There are many studies indicating that elicitation improves pro-health properties of fresh herbs and spices [4,9,10], but there is no information concerning the consumer quality of elicited herbs and the potential of using elicited herbs in food technology in the context of selection of a suitable drying method. Therefore, the aim of the present study was to evaluate changes in some enzymes (polyphenol oxidase and peroxidase activities) and the content of total phenolic compounds, carotenoids, chlorophylls, vitamin C, and microelements, as well as organoleptic quality and color parameters of fresh lovage and its herb dried with various methods and elicited with jasmonic acid and yeast extract.

## 2. Materials and Methods

### 2.1. Plant Material

#### 2.1.1. Growth Conditions

Lovage (*Levisticum officinale* Koch. cv. Elsbetha) leaves growing as a control and elicited with 10 µM of jasmonic acid (JA) and 0.1% yeast extract (YE) were the plant materials used in this study. The growth conditions and method of elicitation were described in our previous manuscript [6].

Lovage seeds (*Levisticum officinale* Koch. cv. Elsbetha) were bought from the Enza Zaden Company. The plants were grown in a growth chamber (SANYO MLR-350H, Sanyo, Japan) at 25 °C/18 °C, photoperiod 16/8 h day/night, with photosynthetic photon flux density (PPFD) at a plant level of 500−700 µmol m^−2^s^−1^, and relative humidity of 75%. The herb seeds were sown into sowing boxes filled with universal soil for sowing seeds. Seven-day-old seedlings were transplanted to 600-mL pots containing universal garden soil with four plants per pot. The plants were watered every other day and fertilized twice (for the first time before plant transplanting and the second time one week after transplanting). Twenty-one-day-old plants were sprayed with a water solution of the tested elicitors (1.5 mL per plant 0.01%). Tween 20 was used as a surfactant.

The concentrations of the elicitors were selected based on our previous study [6], which indicated that 0.1% yeast extract (YE) and 10 µM of jasmonic acid (JA) proved to be the most effective concentrations of the elicitors for lovage plants. Twenty-five days after the elicitation, the plants were collected and next the plant materials were separated into different groups: One fresh group and another group to be subjected to the different drying conditions for analysis Figure 1.

#### 2.1.2. Determination of Plant Yield

Before harvest (Lublin, Poland), the plant height was determined (20 plants measured from each object). After harvest, the herb was weighed, freeze-dried, and weighed again to obtain the total dry weight. The results were expressed as g plant^−1^ [11].

#### 2.1.3. Drying Techniques

Fresh herbs were dried (Lublin, Poland) using four techniques: Natural (traditional) drying (in a darkened room, a temperature of 20 °C to 22 °C, for approximately 7 days), convection drying (in a drying oven at 40 °C for approximately 5 h), microwave drying (in a laboratory microwave dryer, microwave power 360 W at 20 °C, for about 5 min), and freeze-drying (in a lyophilizer at a temperature of −49 °C and pressure of 0.045 mbar) [12,13].

### 2.2. Quality Analysis

#### 2.2.1. Color

The color of fresh and dried lovage leaves was measured using an Envisense NH310 colorimeter. The CIE color values L* (lightness), a* (redness), and b* (yellowness) were measured to describe the color of the lovage leaves. The measuring aperture diameter was 8 mm. The colorimeter was calibrated using the standard of white (L* = 95.82; a* = −0.44; b* = 2.5).

#### 2.2.2. Consumer Quality

The sensory characteristics (color, flavor, appearance) of fresh and dried leaf samples were determined on a scale from 1 to 5 (1—disliked, 5—extremely liked). The panel for the sensory analysis was composed of 20 members aged from 21 to 40 years (12 women, 8 men) [14].

#### 2.2.3. Content of Some Bioactive Compounds

Both fresh and dried plant material was subjected to the same analyses. The weight of analytical samples used for determination of the content of bioactive compounds was equivalent, i.e., the water losses during herb drying were taken into account (the herb samples were weighed before and after the drying process to calculate the water loss). This allowed the comparison of the results expressed per fresh matter (FM) of plant tissue [15].

##### Extraction and Determination of Total Phenolic Compounds in Lovage

Ethanolic extracts were prepared (0.5 g fw or 0.3 g dw in 15 mL of an ethanol/water/hydrochloric acid (70:29:1, *v*/*v*/*v*) solution with sonication at 30 °C for 1 h and then centrifugation at 9000 g for 30 min) for determination of total phenolic compounds in the lovage leaves. The amount of total phenolics was determined using Folin-Ciocalteau reagent [16]. The amount of total phenolics was calculated as gallic acid equivalent (GAE) in mg per g FW (fresh weight).

##### Extraction and Determination of the Levels of Chlorophylls (Chl) and Carotenoids (Car)

Chl and car were analyzed according to the method described by Lin et al. [17]. The contents of Chl *a*, chl *b*, and Car were expressed in mg per g FW.

##### Extraction and Determination of Vitamin C Content

Total vitamin C content was determined as a sum of ascorbic and dehydroascorbic acids according to the methods described earlier by Mazurek et al. [18]. Briefly, 1 g of fresh lovage leaf material was extracted two times with 2 mL of 50% (*w*/*v*) m-phosphoric acid (MPA). The mixture was centrifuged at 16,000× *g*, and extracts were combined and used for further determination. The obtained extract was divided into two parts. One part was used for the determination of ascorbic acid, and the other for determination of the total content of vitamin C after reduction of dehydroascorbic acid. Dehydroascorbic acid was converted to ascorbic acid with 100 mM tris (2-carboxyethyl) phosphine. Quantitative determinations were carried out by calculation of the external standard using calibration curves of the standard and expressed in mg per 100 g FW.

##### Microelement Content

The convectionally dried material was mineralized and the content of elements, i.e., Fe, Cu, Zn, Mn, Ni, Cd, Mg, K, Ca, Na, Pb, Co, and Ag was determined with atomic absorption spectrometry (AAS).

The analyses were carried out at the Central Agroecological Laboratory of the University of Life Sciences in Lublin.

### 2.3. Peroxidase and Polyphenol Oxidase Activities

#### 2.3.1. Polyphenol Oxidase (PPO)

The polyphenol oxidase activity was determined spectrophotometrically based on the procedure reported by Wissemann et al. [19]. First, 0.5 g of plant material was homogenized with 10 mL of 0.1 mol L^−1^ sodium phosphate buffer, pH 6.8, containing 0.01 mol L^−1^ ascorbic acid and 0.5% polyvinylpyrrolidone and extracted for 60 min at 4 °C. Next, the extracts were centrifuged for 20 min (4 °C, 9000× *g*). Catechol was used as a substrate. One unit (1 U) of PPO activity was defined as 0.001 ΔA_420_ per min and was expressed as specific activity in units of enzyme activity per mg protein (U/mg protein).

#### 2.3.2. Guaiacol Peroxidase (POD)

The guaiacol peroxidase activity was measured spectrophotometrically according to the method described by Chance et al. [20]. The extract for POD assays was prepared by homogenizing 0.5 g of the plant material with 10 mL of 0.05 mol L^−1^ acetate buffer, pH 5.6, and shaking for 60 min at 4 °C. Next, the extracts were centrifuged for 20 min (4 °C, 9000× *g*). One unit (1 U) of guaiacol peroxidase activity was defined as 0.001 ΔA_470_ per min and was expressed as specific activity in units of enzyme activity per mg protein (U/mg protein).

### 2.4. Statistical Analysis

All experimental results were means of three independent experiments. The data in the tables and figures represent mean values ± standard deviation (*n* = 9). The results were evaluated for statistical significance using univariate analysis of variance (ANOVA) with Statistica 6.0 software (StatSoft, Inc., Tulsa, OK, USA) and Tukey’s post hoc test. Differences were considered significant at *p* < 0.05.

## 3. Results

The present results showed that the biotic (YE) and abiotic (JA) elicitation generally did not induce statistically significant changes in the growth parameters. Only the jasmonic acid elicitation caused a statistically significant increase in the dry weight of the lovage plants (Table 1).

The consumer quality of lovage was based on three characteristics: Color, flavor, and appearance. The elicitation with JA and YE only slightly affected the sensory quality of the studied herbs. In the case of fresh lovage leaves, the elicited samples (JA and YE samples) were characterized by a better color than the control sample (Table 2). Consumer responses to the dried lovage color acceptability showed that the microwave dried lovage was the least acceptable (the lowest rating was assigned to the CM sample—2.6 ± 1.35), while the freeze-drying, traditional, and convective drying did not affect the assessment of this parameter in comparison to fresh herbs (Table 2). A similar rating was obtained for the overall appearance, with the highest scores for this parameter were recorded for the JA, YEL, and JAT samples (Table 2). No statistically significant differences were observed in the assessment of the smell of the tested lovage, although it should be noted that the highest marks were obtained by the fresh herb samples and those subjected to freeze drying (both controls and elicited samples), as shown in Table 2.

In the case of fresh lovage, the instrumental assessment of the color of the tested herbs showed a statistically significant increase in the L* parameter in the YE sample (which indicates that the YE-elicited lovage leaves were brighter) and the b* parameter (greater proportion of yellow color) in the JA and YE samples compared to the control (Table 2). After drying (with all the techniques employed), both the control and elicited samples were characterized by higher L* and a* values and a lower b* value in comparison to the fresh herbs samples (Table 2). It should be also noted that the largest increase in the value of parameter a* (smaller proportion of green color) was observed in the microwave-dried samples, while the smallest increase in the a* value was noted in the case of the lyophilized samples.

The influence of elicitation on the content of bioactive compounds was determined in fresh and dried lovage leaves (Table 3). In the case of the fresh material, the elicitation with JA caused a statistically significant increase in the content of chlorophylls, vitamin C, and phenolic compounds in comparison to the control (an increase by 49.9%, 17.9%, and 49.6%, respectively). These results were not confirmed, however, in the case of the dried samples. The analysis of the content of vitamin C in the dried lovage demonstrated its highest level in the control samples, regardless of the drying method (Table 3). Comparing the different drying methods, it should be noted that the largest losses of vitamin C occurred in the case of traditional drying, while microwave drying caused the largest losses of chlorophylls and carotenoids in the tested material (Table 3). Noteworthy, in the case of all the drying methods used, the content of chlorophylls and carotenoids was not statistically different between the control and elicited samples (Table 3).

The application of jasmonic acid increased the content of microelements, i.e., Zn, Ni, Cd, Mg, and Na in the lovage leaves by 29.7%, 27.9%, 52.9%, 5%, and 23.2%, respectively. Similarly, higher content of Zn, Ni, Cd, Mg, and Na was determined in the lovage leaves elicited with the yeast extract. However, the biotic elicitation caused a greater increase in the content of these bioelements. In addition, after the induction with the yeast extract, the lovage leaves contained significantly higher levels of Fe and Mn (by 81.5% and 28.5%, respectively) than the control. In contrast, both the abiotic and biotic elicitation caused a decrease in the content of Cu and Ca in the lovage leaves. The application of jasmonic acid also resulted in a decrease in Fe and Mn accumulation in the leaves of the tested herb, while a smaller amount of K was determined in the samples treated with the yeast extract (Table 4).

Peroxidase (POD) and polyphenol oxidase (PPO) specific activities in the control and elicited lovage leaves (fresh and dried with four techniques) were determined (Table 5). In the case of the fresh material, the elicitation caused a statistically significant increase in POD and PPO activities. The treatment with JA resulted in a 98% and 72% increase in the activity of PPO and POD, respectively, while a 54% and 61% increase in these activities (PPO and POD, respectively) was observed after the elicitation with the yeast extract (Table 5). It should be noted that drying caused a significant decrease in the PPO and POD activity in the studied lovage leaves. In the case of dried lovage leaves, the highest activity of PPO material was noted in the microwave dried, whereas the POD activity was similar for all dried lovage leaves (Table 5).

## 4. Discussion

Herb quality from the consumer’s point of view has two main aspects: Pro-health quality and organoleptic quality. Some processes occurring during plant growth or after harvesting can exert an influence on the quality parameters of herbs. As a method of inducing plant secondary metabolism, elicitation may contribute to increased content of some bioactive compounds in plants, including fresh herbs [9,21]. Many previous studies indicated that jasmonic acid and yeast extract are valuable elicitors inducing the biosynthesis of plant pro-health compounds e.g., vitamins, plant pigments, essential oils, or phenolic compounds [5,6,10,22].

In the present study, we also observed that JA elicitation enhanced the synthesis of some bioactive compounds (especially chlorophylls, vitamin C, and phenolic compounds), as shown in Table 3.

The YE-elicited lovage leaves in the present study were characterized only by an increased chlorophyll *b* level (Table 3). These observations indicate that the effect of YE elicitation may depend on the dose of the elicitor used and on plant species. The mechanism of induction of the biosynthesis of these compounds is associated with plant response to stress/elicitors by producing secondary metabolites, including bioactive compounds, by induction of some signal transduction pathway connected with plant systemic resistance [21]. Phenolic compounds are mainly produced via the phenylpropanoid pathway and, as suggested in some studies, their production is focused on the role of phenylalanine ammonia-lyase in the biosynthetic pathway [23]. There are some reports indicating that the application of phytohormones may modulate vitamin C content in plant tissues, but the regulation of its biosynthesis and the mechanisms behind ascorbate homeostasis are still largely unknown [24].

As indicated in a study conducted by Kiczorowska et al. [25], lovage is one of the herbs with the highest abundance of minerals in crude ash in comparison to other herbs, especially Mg and Ca. In our study, the highest K > Ca > Mg content was determined among the examined micro- and macroelements in the lovage leaves (Table 4). These results correspond with the study conducted by the authors of Reference [25]. Many studies have indicated that deficiencies of some mineral nutrients, including Ca, Mg, K, Na, P, Zn, Se, Fe, and Mn, almost certainly impair human health. For example, it is well-known that some micronutrients such as Fe, Cu, Co, Mn, Se, and Zn are beneficial for human health due to their connection with the antioxidant system in the organism, hence their content in some plant foods is very important [26]. Therefore, treatments leading to an increase in the content of some microelements in herbal plants are particularly valuable. As indicated in some studies, stress conditions prevailing during plant growth, as well as the application of elicitors, may affect the mineral composition of herbs [3,26]. In the study conducted by Taha et al. [3], the application of yeast extract (especially at the concentration of 5%, 10%, and 15%) increased the N, P, and K content in neem plants. In the present study, the elicitation with 0.1% yeast extract caused an increase in the Na, Zn, Ni, Cd, Mg, Fe, and Mn content, whereas the application of 10 µM of jasmonic acid resulted in an increase in the Zn, Ni, Cd, Mg, and Na content in studied lovage leaves (Table 4). As suggested by some researchers, the increase in the content of some nutrient elements observed in studied lovage plants may be a result of the positive effect of the tested elicitors, i.e., enhanced vegetative growth and improved uptake of nutrients, as well as production of some phytohormones and conversion of the insoluble form of phosphorous into the soluble one, thereby enhancing phosphorous availability to plants [3].

Drying is regarded as a critical factor for the postharvest quality of herbs. It is a very valuable method for enhancement of the storability of herbs after harvest on the one hand, but may also have a negative impact on the sensory quality and the content of bioactive compounds in the dried material on the other hand. In our study, drying did not generally cause loss of phenolic compounds in the lovage leaves (except for CT and JAT—traditionally dried samples, and JAC—conventionally dried sample). The loss of phenolic compounds during solar-drying (due to the extended drying periods) and oven-drying may be attributed to enzymatic processes driven by polyphenol oxidases, as observed by Mbondo et al. [27]. Unfortunately, drying caused significant losses of vitamin C and plant pigments (carotenoids and chlorophylls) in the studied lovage leaves, and the smallest losses of these bioactive compounds were observed in the freeze-dried samples (Table 3). Vitamin C losses may be connected with the fact that higher temperature and exposure to air or treatments like bruising, cutting, and peeling are the main determinants of vitamin C loss in plant food [28]. These observations correspond with the results obtained by Capecka et al. [15], in which traditional drying resulted in a significant decrease in the contents of vitamin C and carotenoids in peppermint, lemon balm, and oregano. Similarly, a study conducted by Śledź et al. [29] confirmed that drying had a significant impact on chlorophyll degradation in such herbs as basil, lovage, mint, parsley, oregano, and rocket. The loss of carotene may be attributed to the higher rate of isomerization and oxidation at elevated temperature. In turn, as indicated by literature data, retention of this bioactive compound may be significantly improved by reducing the processing time and lowering the temperature [27].

In addition to their health-promoting properties, plant pigments chlorophylls and carotenoids largely shape the color of green plants. In the case of green leafy plants, special attention has been focused on the a* index, which indicates green-red color. In our previous research, elicitation with arachidonic acid and jasmonic acid did not cause significant changes in the color parameters of fresh leafy vegetables like lettuce or endive [30,31]. The present study indicated that fresh lovage leaves elicited with YE were characterized by brighter color, and both the YE-elicited and JA-elicited plants had a greater proportion of yellow color compared to the control (Table 2). However, it is known that treatments in conditions to which natural chlorophylls are sensitive (e.g., extreme pH or temperature), causing degradation of chlorophylls or formation of derivatives of these compounds, will also induce changes in the color, especially in green plants. The impact of drying as one of the herb processing operations on the color of green leafy herbs has been previously studied [29,32]. In a study conducted by Śledź et al. [29], lovage turned out to be characterized by the highest retention of parameter a* during microwave-convective drying in comparison to such herbs as oregano, basil, and mint. As indicated in another study, the color of dried herbs significantly depends on the drying method [33,34]. Microwave and freeze-drying used for drying thyme herb turned out to be the best way in terms of the smallest changes in green color, while convective drying at 60 °C resulted in the largest loss of this color [13]. Our study only partially corresponded with these results, as freeze-drying was the best method for retention of green color in the case of the control and elicited lovage, while microwave drying was found to exert an opposite effect (Table 2). As suggested by the authors of References [13,29], the loss of green color is mainly connected with the loss of chlorophyll during drying, which was confirmed in the present study (Table 2 and Table 3). Additionally, the color changes may be connected with nonenzymatic browning, which is a consequence of Maillard’s reaction accompanying thermal processes during drying [34].

Some enzymes like PPO and POD, which may affect some sensory characteristics of leafy vegetables such as the color, taste, and aroma, can also be involved in determining the quality of herbs. This is related to the fact that the activity of these enzymes is connected with the darkening of plant tissues [35]. The activity of PPO and POD can be influenced by some stress conditions and/or elicitation [36,37]. In the present study, the JA and YE-elicitation of the fresh material caused a statistically significant increase in the POD and PPO activities, whereas a significant decrease in the PPO and POD activities was observed after drying. These results suggest that PPO and POD in the dried material should not affect the sensory properties of the studied herbs (Table 5).

Considering the facts mentioned above and our previous studies [6], which demonstrated that the content of some apterin derivatives (e.g., caffeic acid and apterin ester, as well as sinapic acid and apterin ester) increased in the JA-elicited lovage leaves, thus deteriorating their taste, the present study was also focused on the sensory quality of the elicited herbs. In terms of the sensory quality of the studied herbs, freeze-drying was the best method. However, the sensory evaluation conducted in this study showed no significant differences in the panelists’ opinion about the quality of either the elicited or the untreated lovage samples (Table 2).

## 5. Conclusions

In summary, the elicitation with 10 µM of JA and 0.1% YE had a positive effect on the content of the main groups of bioactive compounds and microelements (probably due to the induction of some signal transduction pathway connected with plant systemic resistance) and did not deteriorate the organoleptic quality of lovage. Nevertheless, it has been observed that the elicited herbs are as sensitive to the loss of bioactive compounds, i.e., vitamin C, as well as chlorophyll and carotenoid pigments as the control plants, but drying does not cause loss of phenolic compounds. The present study indicated that freeze-drying was the best method for drying the studied (both control and elicited) herbs, probably because high temperature was not used in this drying method.

## Figures and Tables

**Figure 1 foods-09-00323-f001:**
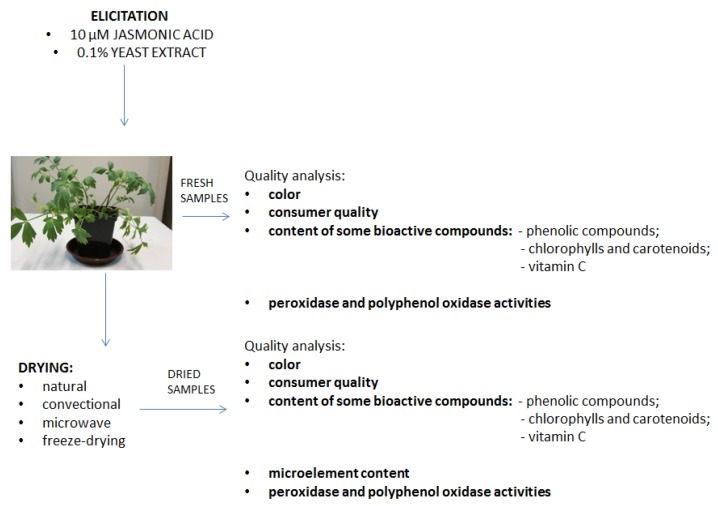
Scheme of the experimental procedure.

**Table 1 foods-09-00323-t001:** Growth parameters of control and elicited with jasmonic acid and yeast extract lovage plants (36-day-old seedlings).

-	Plant Height [cm]	Plant Fresh Weight [g/Plant]	Plant Dry Weight [g/Plant]
C	15.27 ± 0.64 ^a^	1.09 ± 0.35 ^a^	0.17 ± 0.02 ^a^
JA	14.67 ± 1.53 ^a^	1.68 ± 0.32 ^a^	0.25 ± 0.01 ^b^
YE	15.00 ± 1.73 ^a^	1.09 ± 0.02 ^a^	0.16 ± 0.01 ^a^

Abbreviations: C, control; JA, plants elicited with 10 µM of jasmonic acid; YE, plants elicited with 0.1% yeast extract. Mean ± standard deviation. Statistically significant differences (*p* < 0.05) indicated various letters (^a, b^).

**Table 2 foods-09-00323-t002:** Effect of elicitation and drying method on color parameters and consumer quality of lovage.

Sample	Color Parameters			Consumer Quality		
	L*	a*	b*	Colour	Flavor	Appearance
Fresh material						
C	35.57 ± 2.05 ^a^	−7.42 ± 2.61 ^ab^	14.55 ± 1.50 ^c^	3.00 ± 0.00 ^ab^	4.00 ± 0.00 ^a^	4.00 ± 0.00 ^a^
JA	39.20 ± 0.45 ^ab^	−9.26 ± 0.76 ^a^	18.54 ± 3.06 ^d^	4.33 ± 0.58 ^ab^	5.00 ± 0.00 ^a^	4.67 ± 0.58 ^a^
YE	41.97 ± 1.43 ^b^	−5.86 ± 1.48 ^b^	24.28 ± 3.07 ^e^	4.00 ± 0.00 ^ab^	4.67 ± 0.58 ^a^	4.00 ± 0.00 ^a^
Freeze-dried samples						
CL	50.11 ± 0.68 ^cd^	−1.92 ± 0.32 ^cd^	5.58 ± 1.15 ^ab^	4.5 ± 0.71 ^b^	4.2 ± 0.79 ^a^	4.3 ± 0.82 ^a^
JAL	49.90 ± 1.67 ^cd^	−1.22 ± 0.17 ^cde^	4.40 ± 0.42 ^ab^	3.8 ± 0.79 ^ab^	4.2 ± 0.79 ^a^	3.6 ± 1.07 ^a^
YEL	51.83 ± 0.43 ^cd^	−3.14 ± 0.38 ^c^	5.87 ± 0.45 ^ab^	4.4 ± 0.97 ^ab^	4.3 ± 0.95 ^a^	4.6 ± 0.52 ^a^
Microwave-dried samples						
CM	48.55 ± 3.91 ^cd^	0.31 ± 0.08 ^de^	3.24 ± 0.55 ^a^	2.6 ± 1.35 ^a^	3.2 ± 1.55 ^a^	3.1 ± 1.29 ^a^
JAM	54.01 ± 2.85 ^d^	0.71 ± 0.29 ^e^	5.99 ± 0.97 ^ab^	3.6 ± 1.26 ^ab^	3.3 ± 1.06 ^a^	3.5 ± 1.35 ^a^
YEM	52.55 ± 1.10 ^cd^	0.60 ± 0.09 ^e^	5.32 ± 0.10 ^ab^	3.4 ± 1.17 ^ab^	3.1 ± 1.10 ^a^	3.7 ± 0.82 ^a^
Traditionally dried samples						
CT	49.37 ± 0.43 ^cd^	−0.24 ± 0.12 ^de^	6.97 ± 0.16 ^ab^	3.4 ± 1.07 ^ab^	3.4 ± 1.17 ^a^	3.1 ± 1.66 ^a^
JAT	51.77 ± 1.48 ^cd^	0.53 ± 0.01 ^de^	7.75 ± 0.33 ^ab^	4.5 ± 0.53 ^b^	3.6 ± 0.52 ^a^	4.6 ± 0.52 ^a^
YET	47.79 ± 3.60 ^c^	−1.12 ± 0.78 ^cde^	4.79 ± 0.46 ^ab^	3.3 ± 1.16 ^ab^	3.6 ± 0.84 ^a^	3.1 ± 1.37 ^a^
Convectionally dried samples						
CC	50.12 ± 0.20 ^cd^	−1.61 ± 0.30 ^cde^	6.11 ± 0.11 ^ab^	4.2 ± 0.92 ^b^	3.2 ± 0.63 ^a^	3.3 ± 0.82 ^a^
JAC	50.93 ± 0.24 ^cd^	−1.02 ± 0.15 ^cde^	7.20 ± 0.05 ^b^	4.3 ± 0.95 ^b^	3.8 ± 0.79 ^a^	4.1 ± 0.89 ^a^
YEC	51.20 ± 0.65 ^cd^	−0.73 ± 0.15 ^cde^	6.09 ± 0.11 ^ab^	4.1 ± 0.74 ^ab^	3.7 ± 0.67 ^a^	4.4 ± 0.84 ^a^

Abbreviations: C, control (fresh); JA, plants elicited with 10 µM of jasmonic acid (fresh); YE, plants elicited with 0.1% yeast extract (fresh); CL, JAL, YEL—freeze-dried samples; CM, JAM, YEM—microwave-dried samples; CT, JAT, YET—traditionally dried samples; CC, JAC, YEC—convectionally dried samples. Means (±SD) in columns followed by different letters are significantly different (*p* ≤ 0.05).

**Table 3 foods-09-00323-t003:** Effect of elicitation and drying method on the content of bioactive compounds in lovage.

Elicitor	Compounds					
	Chl a (μg/g FW)	Chl b (μg/g FW)	Chl a + b (μg/g FW)	Car (μg/g FW)	Vit C mg/100 g FW	TPC mg/g FW
Fresh material						
C	14.14 ± 1.97 ^b^	5.33 ± 0.43 ^f^	19.47 ± 2.39 ^c^	1.37 ± 0.28 ^c^	49.13 ± 0.63 ^c^	2.40 ± 0.18 ^bcd^
JA	18.00 ± 3.35 ^c^	11.45 ± 0.05 ^h^	29.45 ± 3.35 ^e^	0.89 ± 0.03 ^b^	57.93 ± 0.21 ^d^	4.01 ± 0.31 ^e^
YE	17.07 ± 1.71 ^bc^	6.57 ± 0.64 ^g^	23.64 ± 2.35 ^d^	1.55 ± 0.19 ^c^	49.80 ± 4.62 ^c^	2.74 ± 0.65 ^d^
Freeze-dried samples						
CL	3.33 ± 0.01 ^a^	1.88 ± 0.02 ^e^	5.21 ± 0.03 ^b^	0.33 ± 0.00 ^a^	9.69 ± 0.81 ^b^	2.22 ± 0.12 ^bcd^
JAL	1.81 ± 0.13 ^a^	0.85 ± 0.20 ^bcd^	2.66 ± 0.34 ^ab^	0.18 ± 0.05 ^a^	3.58 ± 0.21 ^ab^	1.87 ± 0.17 ^ab^
YEL	3.17 ± 0.13 ^a^	1.91 ± 0.20 ^e^	5.08 ± 0.33 ^b^	0.25 ± 0.01 ^a^	5.31 ± 0.14 ^ab^	2.32 ± 0.17 ^bcd^
Microwave-dried samples						
CM	1.42 ± 0.20 ^a^	0.39 ± 0.10 ^abc^	1.80 ± 0.30 ^ab^	0.15 ± 0.03 ^a^	6.51 ± 0.75 ^ab^	2.27 ± 0.22 ^bcd^
JAM	0.84 ± 0.01 ^a^	0.16 ± 0.00 ^a^	1.00 ± 0.01 ^a^	0.09 ± 0.00 ^a^	3.52 ± 0.06 ^ab^	1.78 ± 0.23 ^ab^
YEM	1.27 ± 0.09 ^a^	0.25 ± 0.02 ^ab^	1.52 ± 0.12 ^ab^	0.07 ± 0.00 ^a^	5.07 ± 0.77 ^ab^	1.94 ± 0.12 ^abc^
Traditionally dried samples						
CT	2.36 ± 0.07 ^a^	0.95 ± 0.04 ^cd^	3.31 ± 0.11 ^ab^	0.22 ± 0.01 ^a^	4.84 ± 0.11 ^ab^	1.79 ± 0.34 ^ab^
JAT	1.26 ± 0.10 ^a^	0.42 ± 0.05 ^abc^	1.68 ± 0.15 ^ab^	0.14 ± 0.01 ^a^	1.58 ± 0.03 ^ab^	1.28 ± 0.16 ^a^
YET	1.99 ± 0.07 ^a^	0.90 ± 0.04 ^bcd^	2.89 ± 0.11 ^ab^	0.16 ± 0.01 ^a^	1.24 ± 0.08 ^a^	2.66 ± 0.14 ^d^
Convectionally dried samples						
CC	1.84 ± 0.14 ^a^	0.81 ± 0.13 ^abcd^	2.66 ± 0.27 ^ab^	0.23 ± 0.03 ^a^	1.37 ± 0.00 ^ab^	2.55 ± 0.20 ^cd^
JAC	1.12 ± 0.13 ^a^	0.52 ± 0.09 ^abcd^	1.64 ± 0.21 ^ab^	0.14 ± 0.02 ^a^	0.77 ± 0.00 ^a^	1.81 ± 0.17 ^ab^
YEC	2.22 ± 0.07 ^a^	1.16 ± 0.05 ^d^	3.38 ± 0.12 ^ab^	0.23 ± 0.01 ^a^	0.87 ± 0.06 ^a^	2.08 ± 0.27 ^bcd^

Abbreviations: C, control (fresh); JA, plants elicited with 10 µM of jasmonic acid (fresh); YE, plants elicited with 0.1% yeast extract (fresh); CL, JAL, YEL—freeze-dried samples; CM, JAM, YEM—microwave-dried samples; CT, JAT, YET—traditionally dried samples; CC, JAC, YEC—convectionally dried samples. Means (±SD) in columns followed by different letters are significantly different, (*p* ≤ 0.05).

**Table 4 foods-09-00323-t004:** Content of microelements in the control and elicited lovage.

Microelement	Sample		
	C	JA	YE
Fe [mg/kg]	164.67 ± 2.52 ^b^	92.83 ± 3.43 ^a^	299.00 ± 3.61 ^c^
Cu [mg/kg]	8.55 ± 0.79 ^b^	4.66 ± 0.95 ^a^	5.11 ± 0.75 ^a^
Zn [mg/kg]	21.73 ± 0.49 ^a^	28.20 ± 0.56 ^b^	33.60 ± 0.17 ^c^
Mn [mg/kg]	55.40 ± 1.01 ^b^	41.73 ± 1.61 ^a^	71.20 ± 1.18 ^c^
Ni [mg/kg]	9.79 ± 0.46 ^a^	12.53 ± 0.61 ^b^	13.47 ± 0.83 ^b^
Cd [mg/kg]	0.34 ± 0.02 ^a^	0.52 ± 0.04 ^b^	0.83 ± 0.04 ^c^
Mg [g/kg]	2.14 ± 0.02 ^a^	2.25 ± 0.04 ^b^	2.65 ± 0.04 ^c^
K [g/kg]	66.10 ± 3.45 ^b^	68.53 ± 1.75 ^b^	51.07 ± 0.45 ^a^
Ca [g/kg]	27.20 ± 0.17 ^c^	21.10 ± 0.17 ^a^	24.30 ± 0.30 ^b^
Na [mg/kg]	46.13 ± 0.64 ^a^	56.87 ± 0.50 ^b^	58.97 ± 0.67 ^c^
Pb	N.d.	N.d.	N.d.
Co	N.d.	N.d.	N.d.
Ag	N.d.	N.d.	N.d.

Abbreviations: C, control; JA, plants elicited with 10 µM of jasmonic acid; YE, plants elicited with 0.1% yeast extract. Mean ± standard deviation. Statistically significant differences (in rows) (*p* < 0.05) indicated various letters (^a, b, c^). N.d.—not detected.

**Table 5 foods-09-00323-t005:** Effect of elicitation and drying method on the specific activity of enzymes responsible for consumer quality of lovage.

Sample	PPO [U/mg Protein]	POD [U/mg Protein]
Fresh material		
C	102.75 ± 0.60 ^def^	286.53 ± 0.08 ^b^
JA	203.81 ± 19.97 ^h^	494.55 ± 60.73 ^c^
YE	159.26 ± 7.80 ^gh^	461.87 ± 36.00 ^c^
Freeze-dried samples		
CL	17.21 ± 4.28 ^ab^	9.39 ± 4.69 ^a^
JAL	13.77 ± 4.77 ^a^	6.35 ± 2.75 ^a^
YEL	38.25 ± 18.23 ^abc^	5.29 ± 0.01 ^a^
Microwave-dried samples		
CM	88.89 ± 38.49 ^cde^	6.73 ± 0.1 2 ^a^
JAM	144.44 ± 19.25 ^fgh^	5.31 ± 0.75 ^a^
YEM	74.07 ± 32.08 ^bcd^	6.76 ± 0.04 ^a^
Traditionally dried samples		
CT	44.44 ± 19.25 ^abcd^	5.95 ± 0.00 ^a^
JAT	39.22 ± 16.98 ^abc^	9.11 ± 6.31 ^a^
YET	n.d.	19.32 ± 0.00 ^a^
Convectionally dried samples		
CC	30.30 ± 0.00 ^abc^	9.75 ± 3.38 ^a^
JAC	55.56 ± 27.49 ^abcd^	8.23 ± 3.56 ^a^
YEC	58.82 ± 19.61 ^abcd^	11.11 ± 0.00 ^a^

Abbreviations: C, control (fresh); JA, plants elicited with 10 µM of jasmonic acid (fresh); YE, plants elicited with 0.1% yeast extract (fresh); CL, JAL, YEL—freeze-dried samples; CM, JAM, YEM—microwave-dried samples; CT, JAT, YET—traditionally dried samples; CC, JAC, YEC—convectionally dried samples. Means (±SD) in columns followed by different letters are significantly different (*n* = 9; *p* ≤ 0.05).

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
