# Peer review of "Effect of Jasmonic Acid, Yeast Extract Elicitation, and Drying Methods on the Main Bioactive Compounds and Consumer Quality of Lovage (Levisticum officinale Koch)"

_foods, 2020, doi:10.3390/foods9030323_

Round 1

Reviewer 1 Report

Jasmonic acid  induces physiological changes in plants. When it is sprayed after recolection,  it may improve the content of bioactive compounds of certain species.  Based on this premise, this paper is presented, which in principle is shown as an interesting contribution, but which in my opinion needs a reformulation of the structure and presentation of data, which makes my verdict Major Revisions, or even Resubmission.I indicate below the main improvements that should be included in the different sections.

Introduction. It is a little confusing when the bibliography cited as [2] has not been read previously. Please rewrite the text, starting in a much more specific and direct way, referring in more detail to the functions of the jasmonic acid and the processes in which it participates. Some paragraphs of the discussion (vg lines 245 to 254) are more suitable as an introduction than as a discussion. It is good the introduction summarizes all the background and current state of the topic, to set out the objectives that the paper wants to address.

Materials and methods. The structure in so many sub-sections confuses more than help. Points 2.5 and 2.6 are not counterparts to 2.4. And 2.2. & 2.3 are too short. I suggest that you make a figure or graphical abstract to summarize the process in a row. Please do not omit parts that are necessary for understanding the method, and which are described in the quotation [2]

Results. The formatting of the Tables is underworked. They are not attractive, abbreviations confuse more than clarify. It is essential to provide a Table format with much more design, which is intuitvo so that just by looking at the tables we can get a global idea of the paper.that sums up the process.

Discussion. Once all the information that should be in the background has been removed, the discussion paragraphs should be oriented towards an interpretation of the results obtained in paragraph 3, in some detail, or by summing up the ideas keys in a chart or Table.

Conclusions. They are correct, but upgradeable once all the above points have been addressed.

Author Response

Thank you and the Reviewers for helpful comments. All Reviewers' comments were carefully responded and the  manuscript was corrected. Detailed corrections are listed below point by point.

Reviewer #1: Jasmonic acid  induces physiological changes in plants. When it is sprayed after recolection,  it may improve the content of bioactive compounds of certain species.  Based on this premise, this paper is presented, which in principle is shown as an interesting contribution, but which in my opinion needs a reformulation of the structure and presentation of data, which makes my verdict Major Revisions, or even Resubmission.I indicate below the main improvements that should be included in the different sections.

We are very grateful for your detailed review of paper. All suggestions were taken into consideration and the manuscript was corrected point by point.

Introduction. It is a little confusing when the bibliography cited as [2] has not been read previously. Please rewrite the text, starting in a much more specific and direct way, referring in more detail to the functions of the jasmonic acid and the processes in which it participates. Some paragraphs of the discussion (vg lines 245 to 254) are more suitable as an introduction than as a discussion. It is good the introduction summarizes all the background and current state of the topic, to set out the objectives that the paper wants to address.

The Introduction was supplemented with suggested information:

“Jasmonic acid belongs to the group of plant hormones called “jasmonates”. It is a lipid-derived compound synthesized via the octadecanoid pathway. This plant hormone influences some physiological plant process e.g. development, structure, and flowering. Additionally, JA plays an important role in signaling in plants, especially connected with the acquisition of systemic resistance via jasmonate-dependent signaling pathways [2].”

And according to the Reviewer suggestion some paragraphs from discussion has been moved to the introduction:

“Yeast extract is a very interesting natural plant inducer use due to its role in producing some growth regulators as well as its ability to act as a biostimulant of plant growth or the biosynthesis of plant pigments and some other bioactive compounds [3,4]. For example, in a study conducted by Zlotek et al. [5], yeast extract (0.1% and 1%) elicitation increased the level of phenolic compounds and chlorophylls in lettuce leaves. In turn, in marjoram leaves, this elicitor caused an increase in the content of ascorbic acid and chlorophylls [4]. Foliar application of 5% to 20% yeast extract was found to increase the contents of chlorophylls, carotenoids, and phenolic compounds in neem leaves [3]”

Materials and methods. The structure in so many sub-sections confuses more than help. Points 2.5 and 2.6 are not counterparts to 2.4. And 2.2. & 2.3 are too short. I suggest that you make a figure or graphical abstract to summarize the process in a row. Please do not omit parts that are necessary for understanding the method, and which are described in the quotation [2]

The numbering of the subsections has been changed as suggested by the Reviewer and additionally, bolding was used to increase the clarity of the text.

Scheme of experimental procedure has been added – Figure 1:

Please do not omit parts that are necessary for understanding the method, and which are described in the quotation [2]

This information was added:

“Lovage seeds (Levisticum officinale Koch. cv. Elsbetha) were bought in Enza Zaden Company. The plants were grown in a growth chamber (SANYO MLR-350H) at 25/18°C, photoperiod 16/8 h day/night, with photosynthetic photon flux density (PPFD) at a plant level of 500-700 µmol m-2s-1 and a relative humidity of 75%. The herb seeds were sown into sowing boxes filled with universal soil for sowing seeds. Seven-day-old seedlings were transplanted to 600 mL pots containing universal garden soil - four plants per pot. The plants were watered every other day and fertilized twice [for the first time before plant transplanting and the second time one week after transplanting]. Twenty one-day-old plants were sprayed with a water solution of the tested elicitors (1.5 mL per plant 0.01% Tween 20 was used as a surfactant.”

And

“25 days after the elicitation, the plants were collected and next the plant materials were separated into different groups:”

Results. The formatting of the Tables is underworked. They are not attractive, abbreviations confuse more than clarify. It is essential to provide a Table format with much more design, which is intuitvo so that just by looking at the tables we can get a global idea of the paper.that sums up the process.

The formatting of the Tables has been  changed for more clarity of results

Discussion. Once all the information that should be in the background has been removed, the discussion paragraphs should be oriented towards an interpretation of the results obtained in paragraph 3, in some detail, or by summing up the ideas keys in a chart or Table.

The discussion section was rewritten, some information was added:

“The mechanism of induction of the biosynthesis of these compounds is associated with plant response to stress/elicitors by producing secondary metabolites, including bioactive compounds, by induction of some signal transduction pathway connected with plant systemic resistance [21]. Phenolic compounds are mainly produced via the phenylpropanoid pathway and, as suggested in some studies, their production is focused on the role of phenylalanine ammonia-lyase in the biosynthetic pathway [25]. There are some reports indicating that the application of phytohormones may modulate vitamin C content in plant tissues, but the regulation of its biosynthesis and the mechanisms behind ascorbate homeostasis are still largely unknown [26].”

“As suggested by some researchers, the increase in the content of some nutrient elements observed in studied lovage plants may be a result of the positive effect of the tested elicitors, i.e. enhanced vegetative growth and improved uptake of nutrients, as well as production of some phytohormones and convertion of the insoluble form of phosphorous into the soluble one, thereby enhancing phosphorous availability to plants [3].”

“The loss of phenolic compounds during solar-drying (due to the extended drying periods) and oven-drying may be attributed to enzymatic processes driven by polyphenol oxidases, as observed by Mbondo et al. [29] “

“The loss of carotene may be attributed to the higher rate of isomerization and oxidation at elevated temperature. In turn, as indicated by literature data, retention of this bioactive compound may be significantly improved by reducing the processing time and lowering the temperature [29]”

“Additionally, the color changes may be connected with non-enzymatic browning, which is a consequence of Maillard’s reaction accompanying thermal processes during drying [36]”

Conclusions. They are correct, but upgradeable once all the above points have been addressed

Conclusions have been slightly changed:

"In summary, the elicitation with 10 µM JA and 0.1% YE had a positive effect on the content of the main groups of bioactive compounds and microelements (probably due to the induction of some signal transduction pathway connected with plant systemic resistance) and did not deteriorate the organoleptic quality of lovage. Nevertheless, it has been observed that the elicited herbs are as sensitive to the loss of bioactive compounds i.e. vitamin C as well as chlorophyll and carotenoid pigments as the control ones, but drying does not cause loss of phenolic compounds. The present study indicated that freeze-drying was the best method for drying the studied (both control and elicited) herbs, probably because high temperature was not used in this drying method."

In addition, the language correction of the inserted fragments was done.

Reviewer 2 Report

The manuscript studies the changes in the activities of some enzymes, content of some bioactive compounds and the organoleptic quality and color of fresh and dried lovage with various methods and elicited with jasmonic acid and yeast extract.

The results of this work complement the data published by the same authors presented in:

ZÅ‚otek, U.; Szymanowska, U.; Pecio, U.; Kozachok, S.; Jakubczyk, A. Antioxidative and Potentially 339 Anti-inflammatory Activity of Phenolics from Lovage Leaves Levisticum officinale Koch Elicited with 340 Jasmonic Acid and Yeast Extract. Molecules 2019, 24.

The paper is generally well written, and the results are well discussed. However, the authors should address the following aspects:

  • Lines 93-95: the paragraph should be better clarified
  • Line 97 – 15 mL instead of “15 ml”
  • Lines 103-104: Car instead of “car” and Chl instead of “chl”
  • Line 104: mg per g FW instead of “mg per g-1 FW”
  • Lines 123-124 correct the units to 0.1 mol L-1
  • Before Table 1 explicit the days of plant collection
  • Table 1: (p<0.05) instead of “(P<0.05)”

Author Response

Thank you and the Reviewers for helpful comments. All Reviewers' comments were carefully responded and the  manuscript was corrected. Detailed corrections are listed below point by point.

Review #2: The manuscript studies the changes in the activities of some enzymes, content of some bioactive compounds and the organoleptic quality and color of fresh and dried lovage with various methods and elicited with jasmonic acid and yeast extract.

The results of this work complement the data published by the same authors presented in:

ZÅ‚otek, U.; Szymanowska, U.; Pecio, U.; Kozachok, S.; Jakubczyk, A. Antioxidative and Potentially 339 Anti-inflammatory Activity of Phenolics from Lovage Leaves Levisticum officinale Koch Elicited with 340 Jasmonic Acid and Yeast Extract. Molecules 2019, 24.

The paper is generally well written, and the results are well discussed. However, the authors should address the following aspects:

We are very grateful for your review of paper. All suggestions were taken into consideration and the manuscript was corrected point by point:

  • Lines 93-95: the paragraph should be better clarified

Some information has been added:

“Both fresh and dried plant material was subjected to the same analyses. The weight of analytical samples used for determination of the content of bioactive compounds was equivalent, i.e. the water losses during herb drying were taken into account (the herb  samples were weighed before and after the drying process to calculate the water loss). This allowed the comparison of the results expressed per fresh matter (FM) of plant tissue [15]”

  • Line 97 – 15 mL instead of “15 ml” – It was corrected
  • Lines 103-104: Car instead of “car” and Chl instead of “chl” – It was corrected
  • Line 104: mg per g FW instead of “mg per g-1 FW” – It was corrected
  • Lines 123-124 correct the units to 0.1 mol L-1 – It was changed
  • Before Table 1 explicit the days of plant collection – This information has been added
  • Table 1: (p<0.05) instead of “(P<0.05)” – It was changed

In addition, the language correction of the inserted fragments was done.

Round 2

Reviewer 1 Report

I have noticed that all reviewers' comments have been carefully responded and the manuscript has been improved.